# Direct solid-phase synthesis of molecular heterooligonuclear lanthanoid-complexes

Elisabeth Kreidt[1], Wolfgang Leis[1] & Michael Seitz [1 ✉]

Molecular lanthanoid complexes are highly valuable building blocks for a number of important technological applications, e.g. as contrast agents in magnetic resonance imaging (MRI) or as luminescent probes for bioassays. For the next generation of advanced applications based on molecular species, heterooligonuclear lanthanoid complexes with well-defined chemical and structural compositions are required. The great kinetic lability of trivalent lanthanoids so far prevents the realization of such molecular architectures with a universally applicable methodology. Here, we have developed functionalized molecular lanthanoid cryptates as monomeric building blocks which can be directly linked by standard solid-phase peptide synthesis to yield sequence-specific heterooligonuclear lanthanoid complexes. These molecular materials enable unique applications such as the generation of molecular codes with very convenient luminescence read-out.

---

[1] Institute of Inorganic Chemistry, University of Tübingen, Auf der Morgenstelle 18, 72076 Tübingen, Germany. ✉email: michael.seitz@uni-tuebingen.de

One of the great strengths of organic chemistry is its ability to prepare covalently connected oligo- or polymeric structures in a controlled, sequence-specific manner from a set of similar yet different monomeric building blocks, such as amino acids, nucleotides, monosaccharides, etc. A remarkable example is Nature's use of such polymers for the storage of information (e.g., DNA, RNA) or as highly specialized structural and catalytically active materials (e.g., proteins). One of the major breakthroughs in modern synthetic chemistry was the development of artificial chemical methods for the preparation of oligo- and polypeptides, which has revolutionized vast areas of scientific research[1,2]. By combining suitable organic monomers in a highly controlled fashion, a virtually endless chemical and functional space can be created, and many desired properties can be designed. In addition to purely organic systems, metal-based building blocks are highly attractive in this context because of the unique chemical and physical features that these species can provide, e.g., catalytic, magnetic, or photophysical properties. Unfortunately, inorganic coordination chemistry in general does not have a reliable equivalent to the covalent bond C-X (X = C, N, O etc.) of organic chemistry, the latter usually being stable even under challenging conditions. This fact has two important implications in the context discussed here, on one hand the limited stability of the metal complex itself and of the coordinative connection between different metal complex monomers on the other. This usually prohibits the realization of molecular, heterooligonuclear metal complex architectures with a defined and stable composition (e.g., regarding structural integrity, sequence specificity, etc.). Both problems can be circumvented when for the actual connection of the individual metal complex monomers the well-established toolbox of organic synthesis is utilized. The most suitable technique of synthetic organic chemistry is solid-phase peptide synthesis (SPPS), which is well-established, very reliable, and can even be automated. This SPPS approach has been utilized in the past for the construction of oligonuclear transition metal complexes[3–9]. The choice of the metal complex motif is rather limited and usually restricted to stable metal centers with octahedral low-spin-d[6] (e.g., Fe[II], Ru[II], Ir[III]) or square–planar low-spin-d[8] (e.g., Pt[II]) electronic configurations. Prototypical examples of such transition metal-based scaffolds are ferrocenes, octahedral Ru(II)-complexes with pyridine-based ligands, or metallated porphyrins. While the direct SPPS approach has proven viable for certain transition metal building blocks in the past, the important class of molecular lanthanoid complexes has so far not been amenable to this attractive strategy. Molecular lanthanoid complexes have unique magnetic and photophysical properties, and have fascinating applications such as magnetic resonance imaging (MRI) contrast agents, as luminophores in biomedical analysis, or as spectral converter materials[10,11]. Unfortunately, due to their special electronic structure, trivalent lanthanoid cations generally show negligible ligand field stabilization energies and consequently exhibit very labile ligand arrangements and often show only limited thermodynamic and/or kinetic stabilities of their complexes[12]. The harsh conditions during standard SPPS (e.g., very acidic or basic conditions during various deprotection steps) are a considerable challenge for the great majority of known lanthanoid complexes, often leading to decomplexation and for heterometallic systems scrambling of the chemically almost indistinguishable lanthanoids.

Generally, only very few examples of molecular lanthanoid coordination compounds are known which combine several lanthanoid ions in a controlled fashion. Most of the examples are homooligometallic coordination compounds. For example, the multitopic peptide scaffold 1 (Fig. 1) has been realized containing several chelating motifs which can be loaded with several Gd[3+]

cations in a separate step at the end of the synthesis[13]. For the preparation of heterooligonuclear lanthanoid complexes, a much more rigorous control of the covalent assembly process is necessary. Seminal work by Faulkner et al. shows that it is possible to meet these requirements in special cases by connecting several lanthanoid complexes with different functionalizations using covalent transformations such as amide and diazotation coupling chemistry as well as modular, multi-component Ugi reactions[14–19]. This allows for the preparation of heteronuclear lanthanoid complexes such as 2 (Fig. 1) from different building blocks. The composition regarding different lanthanoids in 2 can be controlled by the functionalization of the respective ligand scaffolds. While the realization of species such as 2 is a great achievement, the synthetic methodology involved does not easily lend itself to extending the size and scope beyond a small number of different lanthanoids.

Many known applications could greatly benefit from a general method to combine multiple and different lanthanoid cations in a controlled manner. In addition, such a methodology would enable the development of entirely new avenues for pure and applied science. An especially fascinating one is the preparation of molecular, luminescence-encoded molecules. Analogous to Nature's use of nucleotides for the storage of the genetic code, information can also be encoded in polymeric molecules composed of artificial, luminescent monomers[20]. The use of luminescent monomers would pave the way for completely new applications since the encoded information can very conveniently be read out by straightforward luminescence spectroscopy (Fig. 2). For this type of multiplexing application, trivalent lanthanoids have a unique advantage over traditional organic fluorophores because of the very narrow lanthanoid emission bands, which only have very limited overlap with each other, often allowing for interference-free quantification of every lanthanoid in the presence of other lanthanoids[10,11].

Here, we report a universally applicable methodology for the synthesis of heterooligonuclear lanthanoid architectures using tailor-made lanthanoid monomers and standard solid-phase peptide synthesis to allow for the sequence-specific connection of the building blocks. This provides for a very flexible toolbox for the realization of the full potential of molecular heterooligonuclear lanthanoid complexes. As a prototypical example of the potential of this powerful approach, we show in a proof-of-concept study that molecular materials can be realized that can be encoded by a specific lanthanoid luminescence response.

## Results

**Monomer synthesis and characterization.** For the chemical implementation of this study, we opted for the cryptate-based ligand scaffold 3 (Fig. 3) as the chelating unit for lanthanoid complexation instead of the more commonly used derivatives of DOTA (1,4,7,10-tetraazacyclododecane-1,4,7,10-tetraacetic acid) which also provide rather kinetically inert lanthanoid complexes[21] and are featured in examples 1 and 2 (Fig. 1).

Cryptates such as 3, first introduced by Lehn et al.[22,23] have a number of attractive features that make them very suitable for the purpose of the study outlined. They provide lanthanoid chelates with high apparent stability of the corresponding complexes for the entire lanthanoid series (La–Lu) even under the very acidic and basic conditions required for direct use in SPPS[24]. Furthermore they can be reliably functionalized in the complex periphery[25], allowing for the conjugation to suitable amino acid derivatives. Another very attractive feature of derivatives of 3 is that the aromatic bipyridine chelators can act as efficient sensitizer motifs for the generation of metal-based photoluminescence for a wide variety of lanthanoids, such as Pr, Nd, Sm, Eu,

**Fig. 1 Previous work.** State-of-the-art in oligonuclear lanthanoid complexes.

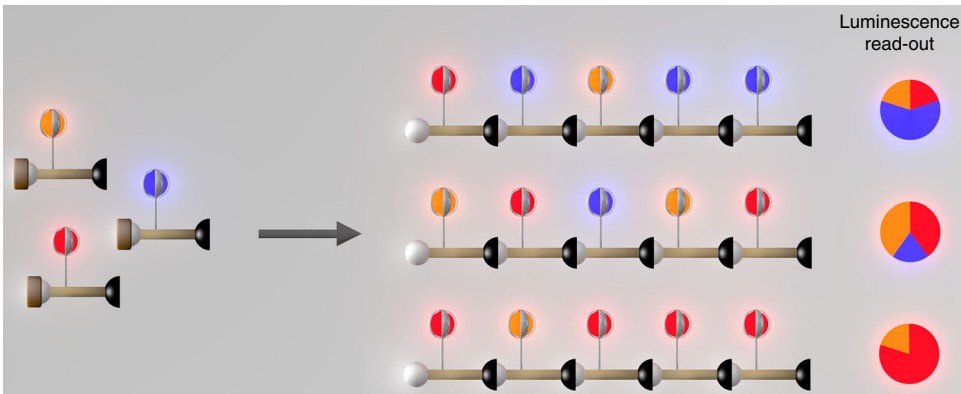

**Fig. 2 Lanthanoid complexes.** Schematic representation of sequence-specific, heterooligonuclear lanthanoid complexes—molecular luminescence nanocodes.

Tb, Er, and Yb[22,25–27]. These properties are the prerequisites for the realization of luminescent nanocodes as outlined above based on these lanthanoid oligomers. After considerable chemical screening for the best conjugation of cryptate scaffold **3** to a suitable amino acid motif, the best candidate was found to be a L-lysine-based building block **Fmoc-Lys(Ln)** (Fig. 3) which has the cryptate complex attached to the ε-amino residue of lysine via a thiourea linkage and which already possesses a Fmoc protecting group at the $N_\alpha$ position, as is most commonly used in standard, Fmoc-based SPPS protocols. Other attempts using a number of peptide coupling methodologies (e.g., via acid chloride or mixed anhydride activation of the carboxylic acid component, via DCC/HOBt or HATU/$^i$Pr$_2$NEt, etc.) with amino- and carboxyl-functionalized cryptate species with the complementary carboxyl- and amino-side-chain-functionalized amino acid building blocks (derived from suitably protected glutamic acid and lysine species) were unsuccessful. For the aim of this study, the corresponding europium, terbium, and samarium cryptate amino acids **Fmoc-Lys(Ln)** were prepared (see Supplementary Methods, Section 1.1 Materials), purified by reversed-phase HPLC (Supplementary Figs. 5–7) and characterized by paramagnetic $^1$H NMR spectroscopy (Supplementary Figs. 11–13). For the purpose of this study, several milligrams of each monomer were found to be sufficient. Scale-up of the reactions performed should be straightforward, and mainly limited by HPLC capacities. Each lanthanoid-

modified amino acid expectedly showed the usual photoluminescence spectrum after excitation in the UV region ($\lambda_{exc} = 305$–320 nm) with the characteristic bands for the respective lanthanoid (Fig. 4; Supplementary Figs. 19–21). In the case of the europium spectrum, the fine-structure of the individual emission bands can give clues to the speciation and the number of different emitting species in solution[28]. Especially, informative is the band $^5D_0 \rightarrow {}^7F_0$ ($\lambda_{em} \approx 575$–580 nm), which only involves non-degenerate energy levels and should therefore only exhibit one single peak for a single species in solution. Closer inspection of this band (Fig. 4) shows multiple components, which is indicative of the presence of more than one species. The observed splitting of the emission band $^5D_0 \rightarrow {}^7F_1$ ($\lambda_{em} \approx 585$–600 nm) also reflects the same conclusion[28]. This phenomenon is very common for cryptates of this type, and is usually connected to the presence of an equilibrium in the inner coordination sphere between different monodentate anions/solvents (e.g., trifluoroacetate vs. CD$_3$OD in this case).

**Solid-phase nanocode synthesis**. From the building blocks **Fmoc-Lys(Ln)**, coupling experiments were performed under standard SPPS conditions, using a protocol starting with a TentaGel R polystyrene resin preloaded with Fmoc-glycine via a Wang-type linker and using the common peptide coupling

**a**

**Fmoc-Lys(Ln)**
(Ln = Sm, Eu, Tb)

· Solv.
(CF₃CHOO)₃

**b**

(Different Ln)

SPPS

**3**

**Fig. 3 Monomeric building blocks and their solid-phase peptide synthesis (SPPS) connection. a** Structure of the lanthanoid building blocks **Fmoc-Lys (Ln)**. **b** SPPS of different monomeric **Fmoc-Lys(Ln)** yielding sequence-specific, heterooligonuclear lanthanoid complexes.

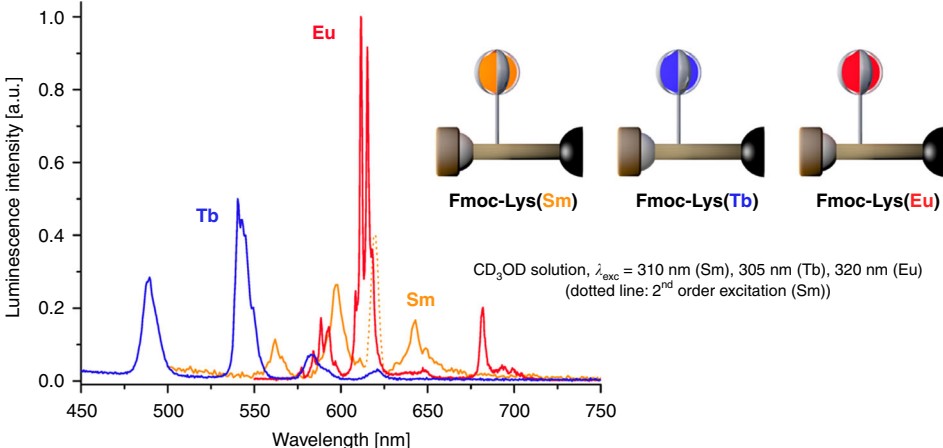

CD₃OD solution, $\lambda_{exc}$ = 310 nm (Sm), 305 nm (Tb), 320 nm (Eu)
(dotted line: 2ⁿᵈ order excitation (Sm))

**Fmoc-Lys(Sm)**   **Fmoc-Lys(Tb)**   **Fmoc-Lys(Eu)**

**Fig. 4 Monomer properties.** Luminescence spectra of **Fmoc-Lys(Ln)** (Ln = Sm, Eu, Tb) exhibiting the characteristic emission bands of each lanthanoid.

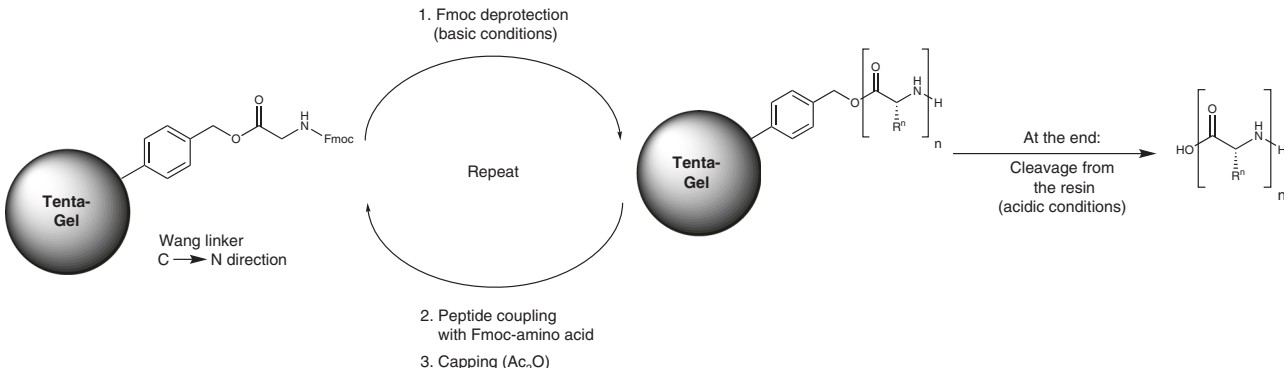

1. Fmoc deprotection
(basic conditions)

Repeat

2. Peptide coupling
with Fmoc-amino acid
3. Capping (Ac₂O)

Wang linker
C → N direction

Tenta-Gel

At the end:
Cleavage from
the resin
(acidic conditions)

**Fig. 5 Nanocode synthesis.** SPPS protocol for the heteronuclear lanthanoid oligomers (Fmoc strategy).

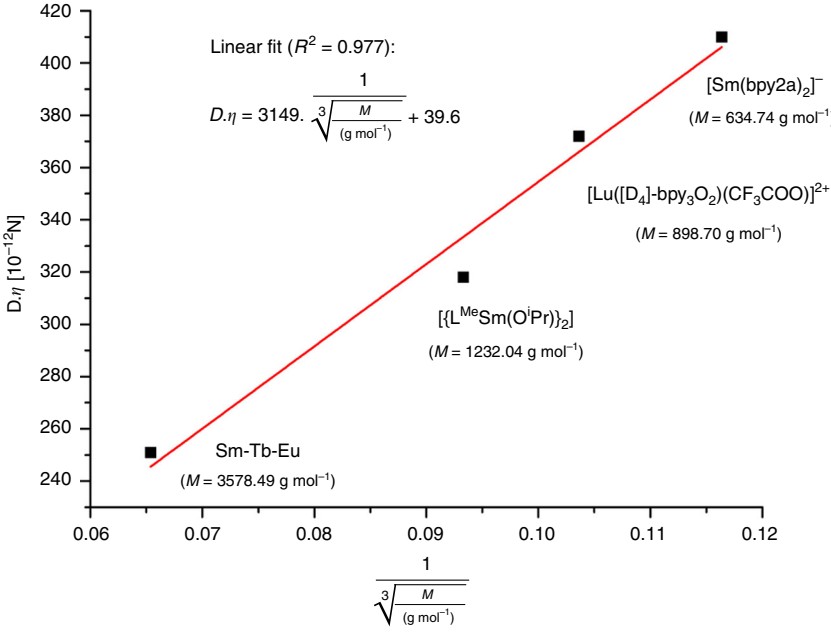

**Fig. 6 Heterotrimetallic Nanocode.** Chemical structure of the nanocode **Sm-Tb-Eu**.

**Fig. 7 Nanocode Identity.** Establishing the size of **Sm-Tb-Eu** by $^1$H DOSY NMR in CD$_3$OD—linear fit of the product of the experimentally determined diffusion coefficients $D$ and the viscosity $\eta$ of the solvents versus $M^{-1/3}$ of the compound **Sm-Tb-Eu** and of reference compounds (see Supplementary Methods, Section 1.6 and Supplementary Fig. 25).

reagent HATU in dimethylformamide. After each peptide coupling, a capping step with acetic anhydride was performed (see Fig. 5).

To increase the flexibility of the growing peptide strand and to minimize electrostatic repulsion, two lanthanoid-based amino acids were separated in the peptide sequence by one interjacent glycine unit. Cleavage of the oligopeptide from the resin was effected by standard acidic conditions (trifluoroacetic acid + 1 vol% water) without the observation of free lanthanoid cations, which would indicate decomplexation. As a proof-of-principle, we used all three different lanthanoid amino acids **Fmoc-Lys(Ln)** (Ln = Sm, Eu, Tb) to prepare a luminescent nanocode on the resin with the peptide sequence **H-Lys(Eu)-Gly-Lys(Tb)-Gly-Lys(Sm)-Gly-OH** (N-terminus → C-terminus) ≡ **Sm-Tb-Eu** with the isotopically labeled $^{13}$C- and $^{15}$N-glycines for analytical purposes (Fig. 6).

**Nanocode characterization and luminescence read out.** The size and identity of the hexapeptide **Sm-Tb-Eu** could not be established reliably by mass spectrometry (MS) because of its high charge (>6+) and the concomitant low concentration in the gas phase under the MS conditions. Instead, diffusion-ordered proton NMR spectroscopy ($^1$H DOSY NMR) in CD$_3$OD (Supplementary Fig. 24) could unambiguously show that the diffusion coefficient for **Sm-Tb-Eu** is fully consistent with the theoretically calculated molar mass of $M = 3.58$ kDa (Fig. 7, see Supplementary Methods, Section 1.6 "$^1$H DOSY NMR Studies").

The structure and composition of **Sm-Tb-Eu** was further corroborated by measuring the photoluminescence spectrum in CD$_3$OD solution. As expected, the characteristic transitions of all three lanthanoids could be identified unambiguously (Fig. 8, see also Supplementary Fig. 22 for comparison with the spectra for

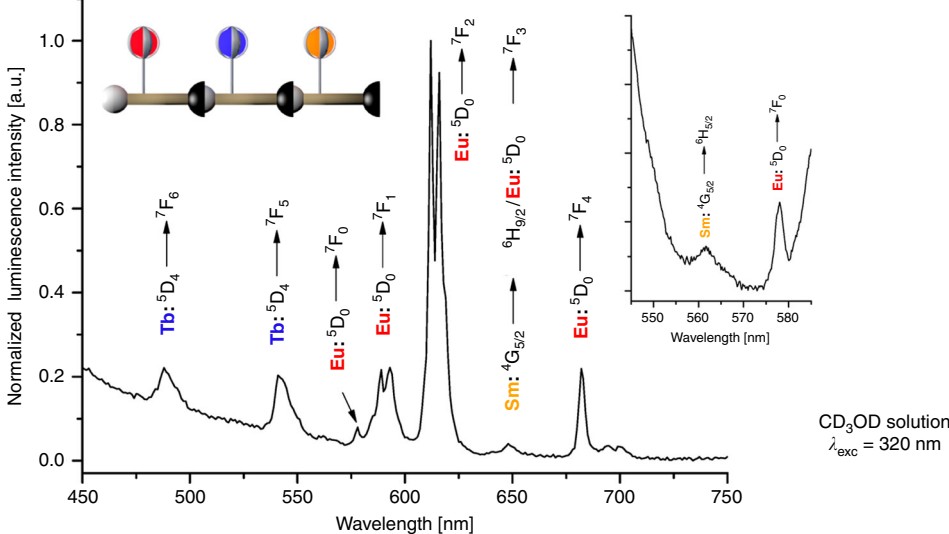

**Fig. 8 Nanocode Luminescence.** Photoluminescence spectrum of **Sm-Tb-Eu** showing the emission bands for all three lanthanoids Sm, Eu, and Tb (inset: zoom-in for the characteristic samarium emission band $^4G_{5/2} \to {}^6H_{5/2}$).

the monomers). The spectrum is dominated by the transitions of the strongly luminescent europium cryptate, but also the transitions of the terbium cryptate can be identified with ease. The samarium bands are considerably smaller, but are still clearly visible, which is expected for this lanthanoid due to its increased sensitivity to multiphonon relaxation via solvent/ligand vibrational modes[29]. While not all spectral regions in the emission spectrum of **Sm-Tb-Eu** are free from partial overlap of bands from two different lanthanoids (e.g., the europium band $^5D_0 \to {}^7F_1$ overlaps with the terbium band $^5D_4 \to {}^7F_4$ around ca. 580 nm), there are certain bands for each of the three lanthanoids that do not suffer from this complication and therefore allow the observation of pure bands (e.g., Sm: $^4G_{5/2} \to {}^6H_{5/2}$ at $\lambda_{em} \approx 562$ nm, Eu: $^5D_0 \to {}^7F_2$ at $\lambda_{em} \approx 620$ nm, and Tb: $^5D_4 \to {}^7F_6$ at $\lambda_{em} \approx 490$ nm; compare also to the individual spectra for the monomers in Fig. 4). The emission spectrum of **Sm-Tb-Eu** proves the presence of all three lanthanoids in the nanocoded material. Together with the results from diffusion-ordered proton NMR spectroscopy, this gives strong evidence for the successful synthesis of the nanocode **Sm-Tb-Eu**.

## Discussion

The lanthanoid complexes **Fmoc-Lys(Ln)** are unique building blocks combining the reliability and stability of the cryptates with the enormous synthetic potential of peptide chemistry. As we were able to show, they are compatible with solid-phase peptide synthesis using well-established standard coupling procedures. This allows for the preparation of heterooligonuclear lanthanoid complexes of potentially any desired sequence, especially with the foreseeable possibility of automating the synthesis with standard SPPS hardware. While the availability of heterooligonuclear lanthanoid architectures had been very limited until now, this methodology will provide easy access to such compounds and will facilitate the exploration of their properties and future innovative applications.

As an illustration of the potentials which come into reach with the SPPS-compatible building blocks **Fmoc-Lys(Ln)**, we prepared the luminescent nanocode **Sm-Tb-Eu**. Such nanocodes have great potential for multiwavelength luminescence coding in molecular compounds, combining the outstanding photophysical properties of the lanthanoids (as already employed in various multiplexing

applications) with the capability of organic chemistry to prepare complex oligo- or polymeric structures in a sequence-specific fashion from simple building blocks (as employed by Nature for the encoding of information in the DNA).

## Data availability

The data that support the findings of this study are available from the corresponding author upon reasonable request.

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

## Acknowledgements

We thank the following institutions for financial support: German Research Foundation (M.S., grant numbers: SE 1448/3-1 Emmy Noether Fellowship; SE 1448/5-1 Heisenberg Fellowship; SE 1448/6-1 Research Grant), German National Academic Foundation ("Studienstiftung des deutschen Volkes", predoctoral fellowship for E.K.).

## Author contributions

M.S. conceived of the work. E.K. executed all synthetic work. W.L. measured and processed all ¹H DOSY spectra. E.K., W.L., and M.S. performed data analysis. E.K. and M.S. wrote the paper, W.L. assisted with editing.

## Competing interests

The authors declare no competing interests.
