## [Peer Review File · Nature Communications]

Reviewers' comments:

Reviewer #1 (Remarks to the Author):

This is a very interesting paper opening up a highly effective approach to heteronuclear complexes derived from the Lehn cryptand, and deploying a mix of synthetic chemistry and spectroscopy to good effect. The work has been carried out well and the conclusions drawn are correct. This work opens up the prospect for encoding with lanthanide complexes: in the system published here, only the lanthanide changes (meaning that the readout will reflect the quantum yield weighted average of the emissive components), but it is easy to conceive changing the ligand as well as the lanthanide, or incorporating sensitising or quenching chromophores into the sequence (which would enable sequence specific readout).

I have no adverse comments, and would be happy to see the manuscript published as it stands. Figure 7 points out key features in the spectrum of the combined array, but it might be worth considering a comment to the effect that (for instance, the Eu 5D0-7F5 transition is partially superposed on the Tb 5D4-7F4 peak)
Steve Faulkner

Reviewer #2 (Remarks to the Author):

Dear Authors

It was with great pleasure that I read your manuscript, and it is with great pleasure that I recommend it for publication.

I have identified several issues, which I recommended i addressed prior to publication. Please find the complete list as comments in the attached pdf.

My major concerns are:

I find that the language used to motivate and frame the work is too strong and that some important references has been neglected.

The speciation of the europium complexes is, according to the emission spectra, diverse. This is not commented on in the manuscript.

And finally, I would appreciate if more of the data used to characterise the molecule is moved from the SI to the body of the manuscript.

Thank you for sharing this beautiful work.

Kind regards
Thomas Just Sørensen

Reviewer #3 (Remarks to the Author):

This is an interesting paper detailing a novel methodology for forming heterooligonuclear lanthanide complexes. This is an unexplored area within lanthanide chemistry as the authors correctly state, with very few multi-lanthanide centre compounds existing in the literature due to inherent kinetic

instabilities. I would recommend publication, with some comments for the authors that should be addressed as part of a revision.

1. the choice of cryptate ligand is explained and has been successful, but did the authors trial other chelate motifs that were not successful? It would be useful to learn if this cryptate ligand can be expanded upon. Indeed, the DOTA motif that is mentioned would be very useful to explore. Virtually all the medically-related Ln complexes feature variations of this DOTA motif so if this chelate unit could also be incorporated into this solid-phase synthesis methodology, the authors will find greater impact for their work.

2. 'considerable chemical screening' is mentioned in terms of chelate conjugation strategy. It would be helpful for readers if these studies can be expanded upon, so others can avoid using unsuccessful strategies and wasting time.

3. In terms of the conjugation strategies, can the authors shorten the linkages between the lanthanide centres, to thereby encourage more chance of through bond and/or through space interactions.

4. I am surprised that Mass Spec evidence could not be obtained for the Sm-Tb-Eu species. What range of MS techniques were attempted? Softer ionisation techniques such as MALDI-Tof would likely succeed.

Responses to Reviewers for Manuscript NCOMMS-19-32361-T

- **Reviewer 1**

- **Partial Superposition of Emission Bands in Figure 7:**

The reviewer suggests that it might be worth mentioning that some emission bands from different lanthanoids overlap in the emission spectrum of the assembled nanocode (Figure 7). We have added a short explanation to this effect that clarifies this issue for the better understanding of the reader.

- **Reviewer 2:**

- **Claims Too Bold / Additional References**

The reviewer states: "I find that the language used to motivate and frame the work is too strong and that some important references has been neglected". We have toned down several passages throughout the manuscript and we have added a number of additional references (see also our responses below to comments from reviewer 2 directly in the manuscript document).

- **Europium Speciation**

The reviewer points out that the emission spectra of the europium complexes show that the speciation is more complex and suggests discussing these features. We have added a short discussion of this aspect and the new reference 28.

- **Moving Data from Supplementary Information (SI) to the Manuscript**

The reviewer suggests that more of the characterization data now in the SI should be moved to the manuscript. Having the very broad readership of Nature Communications in mind, we would prefer to keep the division of the content between manuscript and SI in the current form. In our opinion, it is easier for the non-specialist reader to understand the main points of our manuscript if we do not burden the article with too much detail and instead keep the presentation concise. The primary data that is only necessary for the specialists such as the reviewer or ourselves is in our opinion better documented separately in the SI.

Additional remarks of reviewer 2 were made directly in the pdf version of manuscript and are listed here in the order as they appear in the text.

- **Kinetic Inertness of the Cryptates (Page 1, Abstract):**

The reviewer asks for data to support our description of kinetic inertness of the cryptates that we mentioned in the abstract. This is a very valid point and we do not have solid quantitative data for our new compounds. The problem for the determination of experimental data for this aspect is that this type of cryptate (i.e. 2,2'-bipyridine-*N,N'*-dioxide-based tris(bipyridine) cryptates as used here) has such high, apparent stability (thermodynamic and/or kinetic) that associated complex dissociation equilibria cannot be observed, even under drastic conditions. One of the indirect manifestations is the possibility to perform HPLC on these cryptates. The best illustration of this phenomenon, however, is that the unfunctionalized parent lanthanoid cryptate **3** (see Figure 3) in its enantiomerically pure form can be challenged with 10 equivalents of external lanthanoid(III) salts in refluxing acetonitrile for 5 days and no metal exchange can be observed, which would lead to (partial) racemization (see reference 24 in the revised manuscript for details). The fact that no racemization occurs is ample testament to the complexes' stability. In order to accommodate the valid criticism by the reviewer, we have opted to delete "kinetically inert" from the sentence.

– **Additional Reference (Page 2):**

The reviewer suggests adding a reference. We had already cited the paper in question a little later in the text (reference 14). We would prefer to keep the current position of this reference in the text because it mostly deals with lanthanoid-based multimetallic complex architectures while the paragraph indicated by the reviewer deals with a very general description of the problem.

– **Lanthanoid Complexes vs. Stability Under Harsh Conditions (Page 3, Comment 1):**

The reviewer points out that there are systems by Faulkner et al. where harsh chemistry/purification conditions have also been applied to intact lanthanoid complexes and therefore the sentence in question is too bold. In addition, he suggests the inclusion of Faulkner's work on diazotation chemistry. We intended to convey that the great majority of lanthanoid complexes (mostly with non-DOTA architectures) are rather prone to decomplexation and scrambling under harsh conditions (here most relevant: highly acidic and basic conditions) and we believe that the reviewer would agree with this very general statement. It was not our purpose to give the impression that there are no successful lanthanoid chelates (such as the examples by Faulkner/Sørensen) out there today that can withstand such conditions. In order to make this more obvious, we have rephrased the sentence with the reviewer's comment in mind. In addition, we have now explicitly mentioned the diazotation methodology a few sentences later, where it seemed most appropriate to us, and added the relevant literature precedence as new reference 18 (*J. Am. Chem. Soc.* **2009**, *131*, 9916).

– **Additional Reference (Page 3, Comment 2):**

The reviewer asks to add the same reference as indicated on page 2. Our response is the same as given above (see "Additional Reference (Page 2)").

– **Additional References (Page 3, Comment 3):**

The reviewer asks to add two references. One of them is the same as his two previous comments and has been cited as reference 14. The other request is a new reference and we have gladly added it as new reference 17 (*J. Am. Chem. Soc.* **2004**, *126*, 9490).

– **Current Lack of Generally Applicable Methodology (Page 4, Comment 1):**

The reviewer takes issue with our description that the current state-of-the-art for the synthesis of heteromultinuclear lanthanoid complexes is somewhat limited in general applicability, especially for higher nuclearities. We agree that we should explicitly mention the already established example for the connection by amide linkage and consequently, we have done so in the text and as additional new reference 19. We do disagree, however, that this precedence could be used for the straightforward extension to multinuclear architectures beyond a few lanthanoids. This seems rather difficult to us because so far only monofunctional amine building blocks and monofunctional carboxylic acid building blocks have been coupled. In order to be able to use the full potential of amide-based linkages for the purpose described here, it seems absolutely essential to us to have α -amino acid building blocks configured in a way that allows standard SPPS techniques to be used. The importance of our new strategy is that it is not limited to the connection of three lanthanoid complexes but is in principle only limited by the general scope of SPPS. The genuinely new design of the methodology in which the orthogonality of the protecting groups used in SPPS is combined with the outstanding stability of the lanthanoid cryptates results in an enormous and completely new versatility.

– **Potential Applications of the New Methodology (Page 4, Comment 2):**

The reviewer asks for concrete examples where the newly developed methodology could be used. We agree with the reviewer that the original statement by us "Virtually all known applications would greatly benefit from a general method to combine multiple and different lanthanoid cations in a controlled manner" is probably overselling the applicability a bit. We have rephrased the sentence accordingly in order to tone down this aspect considerably. The paragraph immediately following this sentence, however, seems to us to give the reader a very

good idea what our new compounds could be useful for and we would prefer to leave this part unchanged.

– **Additional Reference (Page 4, Comment 3):**

The reviewer suggests adding a reference to the paragraph in question and we have done so as new reference 20.

– **“Extreme Stability” (Page 5):**

The reviewer takes issue with our description of the lanthanoid cryptates as having “extreme stability” and he is very justified in doing so. We have rephrased the sentence accordingly. The issue concerning stability has already been addressed above (see “Kinetic Inertness of the Cryptates”).

– **Quantities of Fmoc-Lys(Ln) Prepared for this Study (Page 6):**

The reviewer asks for information about the amount of the monomers prepared for this study. A short comment concerning this has been added to the manuscript.

– **Choice of Solvent for the Photophysical Studies (Page 7 and Page 9, Comment 1):**

The reviewer asks why the luminescence spectra of the monomeric lanthanoid complexes were not measured in water. The reason why we used CD₃OD instead for all measurements (including NMR) is fairly straightforward. While Eu and Tb would be emissive enough for the luminescence measurement in water, Sm is one of the less luminescent lanthanoids and its emission is strongly quenched in media with high contents of high-energy oscillators such as O-H-stretching vibrations. In order to get the best luminescence response, we opted for a deuterated solvent of high polarity (important for solubility of the polar compounds) and CD₃OD is simply the most practical in this respect.

– **Cleavage of the Nanocode from the SPPS Resin (Page 8, Comment 1):**

The reviewer asks for data supporting our statements that cleavage of the heterooligonuclear complex was affected “without any decomplexation reactions or other detrimental effects”. We have indirect evidence of the absence of decomplexation by not being able to observe free lanthanoid cations in the cleavage solution. We have rephrased the corresponding sentence in order to clarify this aspect.

– **Mass spectrometry of the monomeric Fmoc-Lys(Ln) (Page 8, Comment 2):**

The reviewer requests mass spectrometric data for the monomeric building blocks. We had already documented this data (MALDI and HR-ESI) in the original version of the SI (see page S8).

– **¹H 1D NMR of the Nanocode Sm-Tb-Eu (Page 9, Comment 2)**

The ¹H NMR of **Sm-Tb-Eu** in CD₃OD is documented as part of the DOSY-NMR studies (see upper part of Figure S23).

– **Overselling Conclusion (Page 10, Comments 1 and 2):**

The reviewer objects to our concluding remarks as overselling our results. We have toned down the conclusions in various respects in order to accommodate the concerns raised.

● **Reviewer 3:**

– **Choice of Chelating Scaffold:**

The reviewer asks if it has been tried to prepare peptides with other ligand scaffolds coordinating the lanthanoid e.g. the DOTA motif. The seminal work by Faulkner has shown without a doubt that DOTA-derived building blocks show enough stability in order to be able to perform covalent linkage chemistry in order to obtain heteromultinuclear lanthanoid complexes. In the context of the intended application (luminescent lanthanoid nanocodes) reported in our present manuscript,

DOTA-related architectures have one drawback and that is the missing antenna function used for the sensitization of lanthanoid luminescence. For many mononuclear DOTA-lanthanoid complexes, this problem has been solved by covalently attaching an antenna moiety suitable for the lanthanoid used. This additional synthetic modification, which usually also requires different antennae for different lanthanoids, would make this approach a little more unwieldy compared to our system. Our tris(bipyridine)-based chelator core has been shown to be able to sensitize a number of lanthanoids in the past which makes our approach relatively straightforward with respect to the sensitization issue. Therefore, we exclusively tried cryptate building blocks so far.

– **Unsuccessful Strategies for Preparation of the Amino Acid-Functionalized Cryptates:**

The reviewer suggests a more detailed description of early unsuccessful experiments towards the preparation of the monomeric building blocks used in the study. The corresponding section of the manuscript has been changed accordingly.

– **Shorter Linkers Between the Lanthanoid Centers:**

The reviewer asks if conjugation experiments with shorter linkers have been performed. Until now this has not been investigated in a systematic manner. The relatively long linker in the current system is simply a consequence of our using of the convenient amino acid lysine for the attachment of the cryptates. We will investigate the reviewer's suggestion in the future.

– **Mass Spectrometry for Sm-Tb-Eu:**

The reviewer asks what range of MS techniques have been tried to obtain mass spectrometry data of **Sm-Tb-Eu**. We have tried the full range of mass spectrometric techniques available to us, including various MALDI and ESI methodologies. For MALDI, a full screening of matrices was performed, equally with no success.